# Gender Differences in the Interplay between Vitamin D and Microbiota in Allergic and Autoimmune Diseases

**DOI:** 10.3390/biomedicines12051023

**Published:** 2024-05-07

**Authors:** Giuseppe Murdaca, Luca Tagliafico, Elena Page, Francesca Paladin, Sebastiano Gangemi

**Affiliations:** 1Department of Internal Medicine, University of Genova, 16132 Genova, Italyelena.kimberly.page@gmail.com (E.P.); 2Allergology and Clinical Immunology Unit, San Bartolomeo Hospital, 19038 Sarzana, Italy; 3Ospedale Policlinico San Martino IRCCS, Largo Rosanna Benzi 10, 16132 Genova, Italy; 4Elderly and Disabeld Department, San Paolo Hospital, 17100 Savona, Italy; 5School and Operative Unit of Allergy and Clinical Immunology, Department of Clinical and Experimental Medicine, University of Messina, 98125 Messina, Italy

**Keywords:** vitamin D, microbiota, allergic diseases, autoimmune diseases, gender, aging, dysbiosis

## Abstract

The synergic role of vitamin D and the intestinal microbiota in the regulation of the immune system has been thoroughly described in the literature. Vitamin D deficiency and intestinal dysbiosis have shown a pathogenetic role in the development of numerous immune-mediated and allergic diseases. The physiological processes underlying aging and sex have proven to be capable of having a negative influence both on vitamin D values and the biodiversity of the microbiome. This leads to a global increase in levels of systemic inflammatory markers, with potential implications for all immune-mediated diseases and allergic conditions. Our review aims to collect and analyze the relationship between vitamin D and the intestinal microbiome with the immune system and the diseases associated with it, emphasizing the effect mediated by sexual hormones and aging.

## 1. Introduction

In the past few years, attention has grown towards the synergic role of the intestinal microbiome and vitamin D in the maturational development of the immune system (IS), as well as their role in the pathogenesis of different immune-mediated and allergic diseases.

Vitamin D exerts its immunomodulatory effect by activating immune cells, such as T and B cells, macrophages, and dendritic cells, as well as by stimulating the production of antimicrobial peptides and neutralizing antibodies. Chronic hypovitaminosis D causes immune paresis, increasing vulnerability to infections and autoimmune disease development [1].

The intestinal microbiota plays a role in the regulation of IS also through the synthesis of some metabolites, including short-chain fatty acids (SCFA) such as butyric acid, propionic acid, and acetic acid, which have properties affecting the development and function of the intestinal IS [2]. These metabolites are able to increase the barrier function of the intestinal epithelium and the degree of immune tolerance [3]. Dysbiotic alterations could lead to a greater predisposition to the development of immune-mediated diseases and increase the risk of infectious events [4].

In an era of growing evidence regarding the existence of gender differences in terms of disease prevalence, we aim to highlight the effect that sexual hormones have on vitamin D and on the composition of the microbiome, as well as the consequent effects on the presentation of autoimmune and allergic diseases.

The aging process, as well as sex differences, is able to negatively influence vitamin D values and the biodiversity of the microbiome, resulting in an increase in the levels of markers of systemic inflammation with potential repercussions for all immune-mediated diseases.

The following paragraphs will give a summary of the characteristics of vitamin D and the gut microbiome in their relationship to the immune system and associated diseases, taking into account age and sex as relevant mediators in this regard.

## 2. Materials and Methods

The aim of our work was to analyze the relationship between vitamin D and the intestinal microbiome with the immune system and the pathologies associated with it, with particular reference to the effect mediated by sexual hormones and the physiological aging process.

For this purpose, we conducted a systematic review of the literature published on the MEDLINE database. Our research did not include limitations regarding publication date as long as the manuscripts were freely accessible in their complete form.

The keywords we used in our search included the terms “vitamin D”, “microbiota”, “allergic diseases”, “autoimmune diseases”, “aging”, and “dysbiosis”.

We collected 126 articles regarding the topics of interest, which we have critically analyzed in order to provide a summary overview of the current knowledge on the topic concerned.

## 3. Synthesis and Mechanism of Activity of Vitamin D

According to scientific evidence, vitamin D has acquired over time an increasingly important role in human physiology and pathophysiology [5]. This is thanks to its relevant role as an actual hormone, implied not only in bone metabolism but also in immunomodulation [5].

Vitamin D exists in two main forms: vitamin D2 (ergocalciferol) and vitamin D3 (cholecalciferol). While vitamin D2 can be obtained mainly from a daily diet, vitamin D3 can also be obtained from a diet but is also synthesized at the skin level from its precursor, 7-dehydrocholesterol, after exposure to adequate ultraviolet B radiation (UVB) [6]. Ultraviolet radiation, while playing a relevant role in vitamin D activation, also acts directly on IS, especially at the skin level [7]. Pre-D3 is thermodynamically unstable and subsequently undergoes a thermal isomerization with D3 [8]. Once in our body, it has to undergo two consecutive hydroxylations. Initial hydroxylation occurs in the liver with several enzymes (particularly cytochrome P450 2R1) and the production of 25-hydroxyvitamin D (25OHD) [9]. A second hydroxylation, resulting in the production of 1,25-dihydroxycholecalciferol [1,25(OH)2D], occurs mainly in the kidney by the cytochrome p450 27B1 (CYP27B1) enzyme as well as other tissues, including some cells of the IS [9]. At the level of immune system cells, CYP27B1 is a relevant player, and its activity is strongly regulated by cytokines and interleukins, where vitamin D activation appears to play an intracrine or paracrine role [10]. The biologically active form of vitamin D is 1,25(OH)2D, and its production is highly regulated by several factors, including hormones such as parathyroid hormone (PTH) and fibroblast growth factor 23 (FGF23) [9]. Specifically, while PTH increases the production of 1,25(OH)2D, FGF23 has the opposite effect [9]. As already highlighted in part, it is interesting to note that its synthesis can also be stimulated by certain cytokines such as tumor necrosis factor alpha (TNFα) and interferon gamma (IFN-γ), namely at the level of keratinocytes and macrophages [9]. Furthermore, vitamin D3 can undergo side chain hydroxylation by cytochrome P450 Family 11 Subfamily A Member 1 (CYP11A1), resulting in the production of various metabolites [11]. CYP11A1 is also expressed in immune system cells such as T and B cells and monocytes, and its metabolites appear to have multiple immunomodulatory effects [11,12]. Upon conclusion of vitamin D metabolism, both 25OHD and 1,25(OH)2D are catabolized by Cytochrome P450 Family 24 Subfamily A Member 1 (CYP24A1), resulting in 24-hydroxylated products [13].

The biological activity of 1,25(OH)2D is mediated mainly through the vitamin D receptor (VDR) [14]. VDR is a member of the steroid receptor family and fulfills its function as an obligate heterodimer in association with the retinoid X receptor (RXR) [14]. The 1,25(OH)2D acts on VDR by inducing a conformational change that promotes the interaction with RXR and coregulatory complexes [14]. Activation of the VDR/RXR complex also involves genome-wide action, specifically on DNA sequences called vitamin D response elements (VDREs) [14]. Other factors are also involved in mediating the metabolic effects of vitamin D beyond VDR. A relevant role is played by RAR-related orphan receptor alpha (RORα) and gamma (RORγ) [15].In particular, 1,25(OH)2D and several metabolites derived from the hydroxylation of vitamin D by CYP11A1 are capable of inhibiting interleukin 17 (IL-17) through RORα and RORγ [15,16].

The so-called canonical actions of vitamin D are mainly related to calcium and phosphate metabolism: 1,25(OH)2D acts at the intestinal level by promoting the absorption of calcium from the daily diet and at the renal level in the distal tubule by easing its reabsorption [14]. However, this action is twofold at the bone level. In a negative calcium balance, PTH and 1,25(OH)2D promote its reabsorption and reduce bone matrix mineralization. In a normal or positive balance, 1,25(OH)2D promotes osteoclastogenesis by stimulating osteoprogenitors, while in mature osteoblasts it has both an anticatabolic and anabolic action [14]. It is important to highlight that 1,25(OH)2D has a feedback mechanism with PTH and FGF23, inhibiting the production of PTH and stimulating the production of FGF23 [17].

Other vitamin D functions have been highlighted in various systems and apparatuses, with increasing relevance. Vitamin D’s canonical mode of action is related to calcium and phosphorus metabolism, while its other actions include the cardiovascular system, the oncological domain, and the IS. In our review, we will focus on the relationship between vitamin D’s effects and IS.

## 4. Physiological Composition of the Gut Microbiome and Dysbiosis

The intestinal microbiome, or “commensal”, is represented by the entire microbial community populating the gastrointestinal (GI) tract of mammals, whose greatest concentration is located in the colon [18]. The human gut microbiome can be represented by 3.8 × 10^13^ microbes in a standard adult male. The human colon gut microbiota is made of five main phyla: Firmicutes, Bacteroidetes, Actinobacteria, Proteobacteria, and Verrucomicrobe, of which the two dominant phyla are Firmicutes and Bacteroidetes, which account for 90% of the total [19]. Some differences were observed in the composition of the microbiome between the ascending and descending colon, with the Clostridiaceae, Lachnospiraceae, and Bacteroidaceae families dominating the microbiome structure at this level [20].

Despite the high interindividual variability in the composition of the gut microbiome, a central type has been identified and is responsible for a number of physiological and homeostatic functions, such as digestion of polysaccharides, development of the IS, defense against infections, synthesis of vitamins, storage of fats, regulation of angiogenesis, and the development of behavior [21].

The shift of the intestinal microbiome from a normal condition, known as “eubiosis”, to an altered condition of its composition and function is called “dysbiosis” [22]. Dysbiosis can be described as both a reduction in species diversity and a change in the relative abundance of “healthy” versus “less healthy” microbes. Considering the significant variability of the term, some specific indices have been proposed, but the most commonly used one is the Microbiome Health Index (MHI), which considers the bacterial distributions within the Firmicutes, Bacteroidetes, and Proteobacteria phyla [23].

Numerous recent studies suggest that the condition of dysbiosis may be implicated in the etiology of many diseases, including IBD, irritable bowel syndrome (IBS), diabetes, obesity, and cancer [24,25,26]. Examples of the likely association between intestinal dysbiosis and some human diseases can be found in the reduction in bacterial species with anti-inflammatory properties, such as Faecalibacterium prausnitzii, or in the presence of potentially pro-inflammatory bacteria such as Bacteroides and Ruminococcus gnavus, which can be associated with IBD [27].

## 5. Vitamin D and the Gut Microbiome in Immunomodulation

As indicated previously, 1,25(OH)2D has several roles at the level of the IS, both regarding cells of innate and adaptive immunity. One of its functions on the innate IS is stimulating the production of pattern recognition receptors (PRRs), antimicrobial peptides such as cathelicidins, alpha- and beta-defensins, and cytokines in both myeloid and epithelial cells [28]. The activation of toll-like receptors (TLRs), particularly TLR2, by microorganisms increases the expression of CYP27B1 in keratinocytes, which in turn increases levels of 1,25(OH)2D in the presence of adequate amounts of 25OHD and can stimulate cathelicidin expression [29]. Furthermore, 1,25(OH)2D induces in macrophages the production of reactive oxygen species in support of their anti-microbial activity [30]. It also describes how 1,25(OH)2D promotes monocyte-macrophage differentiation, reduces their production of proinflammatory cytokines and chemokines, inhibits the expression of innate immunity receptors (like TLR2, TLR4, and TLR9), and downregulates the expression of major histocompatibility complex (MHC) class 2 [31].

This bidirectional action of vitamin D on the innate IS also occurs in innate lymphoid cells (ILCs). While in some scientific works, vitamin D is found to be important for the cytokine response of ILCs to infectious processes [32], in others it has an anti-inflammatory or antagonistic effect in this sense [33,34].

What was previously expressed highlights how vitamin D plays a key role in controlling the microbiome through innate IS, especially in the gut, and also in preventing bacterial translocation into the bloodstream [35].

Vitamin D has been shown to be relevant to the activity of natural killer (NK) cells in different clinical contexts [36,37] and this occurs due to vitamin-induced up-regulation of cytotoxicity receptors as well as down-regulation of killer inhibitory receptors [38]. At the same time, vitamin D could induce a regulatory profile in this cell line through the production of interleukin 10 (IL-10) [39].

The overall literature shows a negative effect of vitamin D on dendritic cells (DCs), with obvious repercussions on the adaptive IS as well [9]. There are many articles showing that 1,25(OH)2D affects DCs by altering and reducing the maturation of medullary precursors in this cell line, as well as reducing the expression of MHC II, CD86, and CD80 and increasing C-C chemokine receptor type 5 (CCR5), DEC205, F4/80, and CD40 [40,41]. It is described in the literature how vitamin D leads DCs to a more tolerogenic phenotype with the conversion from CD4 T cells to IL-10-secreting regulatory T cells (Treg) [42].

With innate IS, the overall effect of vitamin D is activation, with its anti-inflammatory effects in specific cell lines described in some scientific articles. It is different for the adaptive IS, where vitamin D seems to have an overall anti-inflammatory effect on all its components [43].

In T lymphocytes, 1,25(OH)2D has been shown to induce a T helper (Th) cell polarization by inhibiting Th1 and Th17 and increasing Th2 development through both an effect mediated by antigen presenting cells (APCs) and also a direct effect [44,45]. As partly highlighted earlier, 1,25(OH)2D also leads to an increased presence of Treg cells, particularly positive for forkhead box P3 (Foxp3) and CD4 [41,46].

Finally, in relation to B lymphocytes, 1,25(OH)2D reduces their proliferation as well as differentiation into plasma cells and post-switch memory B cells [47]. This obviously also has an impact on their immunoglobulin production.

Also, according to the current literature, the microbiota has a close link with the IS. This link is created during a critical time window in the first years of life, resulting in long-lasting impacts on multiple IS components that contribute to homeostasis and susceptibility to infectious and inflammatory diseases [48]. The most studied interface for host-microbiota interactions is the intestinal mucosa. A characteristic of intestinal IS is its ability to establish immune tolerance to a wealth of harmless microorganisms while simultaneously preserving immune responses against pathogenic infection [49].

The intestinal microbiota synthesizes several metabolites deriving from undigested exogenous food components that reach the colon, as well as endogenous compounds generated by microorganisms and the host. These microbial metabolic products come into contact and interact with host cells through the layer of epithelial cells that constitute the mucosal interface between the host and microorganisms, thus influencing their immune responses [50].

Among the diverse metabolites synthesized by the gut microbiome, SCFAs, such as butyric acid, propionic acid, and acetic acid, have been shown to have broad effects on the development and function of the gut’s IS [51].

SCFAs bind G protein-coupled receptors (GPCRs), such as GPR41, GPR43, and GPR109A, on the surface of epithelial cells and immune cells. Diffusion of SCFAs into host cells results in their metabolism and/or inhibition of histone deacetylase (HDAC) activity [48]. The effects of SCFAs are multiple and include increased epithelial barrier function and immune tolerance, through specific mechanisms: increased mucus production by intestinal goblet cells; inhibition of nuclear factor-κB (NF-κB); activation of inflammasomes and subsequent production of IL-18 and downstream antimicrobial peptides; increased secretion of secretory IgA (sIgA) by B cells; reduced expression of T cell-activating molecules on antigen-presenting cells, such as DCs; and increased numbers and function of colonic Treg cells, including their expression of FOXP3 and their production of anti-inflammatory cytokines [transforming growth factor-β (TGFβ) and IL-10] [52,53,54]. The immunomodulatory effect of SCFAs also occurs at the level of neutrophils, through which they induce chemotaxis through binding to GPR43 and subsequent activation of p38 MAPK [55].

## 6. Role of Vitamin D and the Gut Microbiome in Autoimmune and Allergic Diseases

As we inferred in the previous paragraph, vitamin D and microbiota have a significant impact in an immunomodulatory sense. This is also associated with an important role in disease processes associated with the IS, particularly autoimmune and allergic diseases [56,57].

Starting with vitamin D and, in particular, autoimmune processes, there are several pieces of evidence showing that hypovitaminosis D may be a risk factor associated with the development of autoimmune diseases in genetically predisposed individuals [1,58]. This is associated with an altered self-tolerance of the IS in different cellular populations, where the several processes induced by vitamin D, as seen above, also play a role [58]. Vitamin D deficiency can not only be a risk factor for the development of these diseases but, especially if severe, it can be an aggravating factor in their symptomatology [1].

There are also several clinical studies showing that adequate vitamin D supplementation in the population can lead to a reduced incidence of autoimmune diseases [59]. The work of Sizheng Steven Zhao et al. [60] showed through a Mendelian randomization how variants within gene regions implicated in vitamin D biology are associated with the risk of developing psoriasis and systemic lupus erythematosus.

At the genetic level, certain single nucleotide polymorphisms within the VDR gene (particularly ApaI, BsmI, TaqI, and FokI) have demonstrated associations with susceptibility to autoimmune diseases [61]. These changes in vitamin D-associated enzymes and receptor complexes have also raised the hypothesis of a possible association between resistance to this molecule and autoimmune diseases [16]. This would imply that in these specific cases, supplementation with high doses of vitamin D may be the optimal strategy both in terms of prevention and overall improvement of the symptom picture in patients with these diseases [16].

There are numerous autoimmune diseases that are associated with vitamin D in their pathophysiological processes but, in particular, those best described in the literature are autoimmune thyroid diseases, rheumatoid arthritis (RA), inflammatory bowel diseases (IBD), and systemic lupus erythematosus (SLE) [28,56,62].

There are also data showing an association with the development of allergic diseases. Primarily, some articles show the association between ultraviolet radiation exposure and the development of allergic diseases, with important implications for the incidence of these pathologies also depending on latitude [63]. In particular, it has been seen that vitamin deficiency in both mother and infant increases the risk of developing food allergies, particularly in the second year of life [64]. With regard to allergic diseases of the airways, there are reduced vitamin D levels associated with increased IL-31 and IL-33 [65]. There is also evidence of improvement in the symptomatic condition when vitamin D supplementation is given, in particular regarding asthma [66]. Significantly lower levels of vitamin D were found mainly in the pediatric population with atopic dermatitis [66]. Moreover, its supplementation with a weighted average dose of 1500–1600 IU for up to 3 months showed clinically relevant improvements in these patients [67].

VDR receptor polymorphisms also appear in today’s literature to have significance in this context. In particular, VDR rs7975232 ApaI, rs1544410 BsmI, and rs731236 TaqI gene polymorphisms seem to be the most important polymorphisms in this regard, with variations related to both specific allergic pathology and ethnicity [68].

There is also evidence showing the gut microbiota’s possible role in immune-mediated diseases. Intestinal dysbiosis is closely involved in the etiopathogenesis of various extra-intestinal autoimmune diseases. This is the case, for example, of type 1 diabetes, the onset of which is frequently accompanied by a decrease in specific taxa such as Bifidobacterium, Lactobacilli, Bacteroides, Faecalibacterium, and Akkermansia [69]. These bacterial species are in fact involved in the regulation of glucose metabolism and responsible for specific mechanisms such as glucagon synthesis, the production of insulin receptor substrate, the suppression of hyperglycemia, and the production of the GLP-1 hormone, which is involved in insulin’s synthesis, secretion, and sensitivity [70,71].

The reduction in intestinal bacterial varieties is also involved in the development of other autoimmune diseases, such as SLE, in which a significantly lower ratio of Firmicutes/Bacteroidetes has been observed [72], where Firmicutes were inversely correlated with the disease activity index [73]. In patients with RA, there is typically an increase in Prevotella covers, responsible for the synthesis of the Pc-p27 protein, which could trigger a Th1-mediated immune response by binding to HLA-DR [74].

Another important role appears to be played by SCFAs, which have the ability to reach other organs from the gut, such as the brain and lungs. Here, they act directly or indirectly on local or resident antigen-presenting cells to reduce inflammatory responses associated with neuroinflammation and allergic tract disease, respectively [75]. In support of this hypothesis, it is highlighted that in patients suffering from rhinitis, atopic eczema, asthma, and food allergies, alterations of the intestinal microbiota and reduced levels of Tregs are frequently found [76,77]. In the genesis of allergic diseases, the intestinal microbiome has been shown to play a major role. It is in this scenario that the hygiene hypothesis has been theorized, with which it is assumed that an immune deviation in the reactions of Th2 cells occurs due to reduced early microbial exposure and the lack of fecal microbial diversity [78]. Bacterial colonization of the intestine is fundamental to the differentiation of Th cells into Th1, Th2, Tregs, and Th17 cells [79]. Furthermore, intestinal bacteria such as Lactobacillus, Bifidobacterium, Bacteroides, and Clostridium, as well as their metabolites, induce peripheral Treg cells that control enteric and pulmonary inflammation [80,81].

Refer to Figure 1 for a graphical representation of the possible pattern of interaction between hypovitaminosis D, dysbiosis, and alterations of the immune system that can lead to autoimmune or allergic diseases.

## 7. Sex and Age Differences in the Role of Vitamin D and the Gut Microbiome in Immune-Mediated Processes

In this review, we want to highlight the role of different genders and ages in the complex mechanisms described in the previous paragraph.

There is a bidirectional relationship between estrogen and vitamin D [82]: different levels of estrogen decrease the expression of CYP24A1, involved in the inactivation of vitamin D, and increases the expression of the VDR gene [82]. Concurrently, vitamin D has been shown to down-regulate the expression of aromatase, which converts testosterone to estrogen, in immune cells [82].

This interaction between vitamin D and sex is also expressed in the context of different immune-mediated diseases, which tend to be more incidental and prevalent in females, especially autoimmune diseases. Literature shows that in patients with SLE, vitamin D exhibits its immunomodulatory effects at low concentrations of oestradiol but not at higher concentrations [83]. Another example of this interplay between vitamin D and estrogen is RA, where synovial fluid is characterized by an elevated level of estrogen due to increased aromatase activity at this level [84]. Subsequently, estrogen is converted into pro-proliferative and pro-inflammatory 16-hydroxylated estrogens [84]. At this level, there is vitamin D, which can down-regulate aromatases [84].

Sex differences and the relationship between vitamin D and allergic diseases are investigated in the literature. In a pediatric setting, vitamin D levels have been shown to correlate inversely with IgE levels, but only in females [85]. At the same time, lower concentrations of vitamin D have been observed in males with at least two allergies compared to males with only one allergy or females with at least two allergies [85].

In the last few years, scientific research has identified significant differences between the two different sexes in terms of gut microbiome composition and the abundance of specific taxa. These gender differences in the gut microbiome composition are defined by the role that sex hormones play in it.

The microbiome can influence estrogen levels, which can in turn be influenced by the composition and diversity of the microbiome itself through the expression of B-glucuronidase (an enzyme capable of metabolizing estrogens) [86,87]. Similarly, intestinal commensal bacteria belonging to the genus Clostridium scindens are capable of coding hydroxysteroid dehydrogenases and other enzymes involved in the conversion of glucocorticoids into androgens [88,89].

It has emerged from various studies that higher levels of serum hormones were associated with greater intestinal microbial diversity compared to those with lower hormone levels in both sexes, indicating the role that sex hormones play in the composition of the intestinal microbiota [90]. Studies conducted on the gut microbiome of women on estrogen-progestin hormone therapy found that they had lower alpha diversity, distinct beta diversity, and differentiated gut microbial metabolic pathways compared to women who did not use oral contraceptives [91]. The bacterial genres that have been proven to be significantly related to higher levels of testosterone in men were Acinetobacter, Dorea, Ruminococcus, and Megamonas, while Slackia and Butyricimonas were the ones more correlated with higher levels of estradiol in women [92,93,94].

Sex differences in the intestinal microbiota are reflected in the different development of certain diseases between the two sexes [95], e.g., occurring twice as often in females than males of IBS [96]. Garcia-Fernandez et al. [97] discovered that specific alterations in the intestinal microbiota are correlated with the development of cardiovascular diseases, as, in their study in the male gender, there was a decrease in the Barnesiellaceae family, while there was an increase in the Ruminococcaceae, Bilophila, and Phascolarctobacterium genres in females.

Studies conducted concerning the development of obesity and metabolic diseases have identified a slower increase in the Firmicutes/Bacteroidetes ratio in female mice. Differences in the abundance of specific genres in patients with metabolic syndrome have been observed: Veillonella, Methanobrevibacter, Acidaminococcus, Clostridium, Roseburia, and Faecalibacterium in males; Bilophila, Ruminococcus, and Bacteroides in females [90].

On the other hand, we know that with aging, there is often an increased risk of hypovitaminosis D. In the literature, there are several explanations for this phenomenon. It is well known that there is a reduction in the production and metabolism of vitamin D at the level of the skin [98,99,100]. However, it is worth considering that with aging, the reduction in vitamin D production occurs in a context in which the skin nevertheless remains capable of producing vitamin D with adequate UVB exposure [100]. Herein lies the second concomitant cause of hypovitaminosis D in older adult patients, which is the reduction in sunlight exposure, especially in institutionalized adults [101]. It is also commonly described that aging is associated with a reduction in sex hormones, which relate both to vitamin D and IS [102].

Changes in the IS developed during aging are described in the literature as “inflammageing” and immunosenescence [103]. Inflammageing is characterized by the development of chronic low-grade inflammation, while immunosenescence is characterized by the reduced effectiveness of the IS to respond to different pathogens as well as cancer [103]. From a pathophysiological point of view, in older adults with vitamin D deficiency, there are increased levels of markers of inflammation that can potentially spill over into immune-mediated diseases [104]. This is the case of what appears to be happening in RA, where in malnourished subjects with reduced levels of vitamin D there is an increased level of inflammation as well as an increase in disease activity [105].

The relationship between aging, vitamin D, and allergic diseases appears to be less investigated; however, the overall negative effect of vitamin D deficiency on diseases such as asthma seems to be confirmed in older adult patients [106].

Similarly, as for vitamin D, the intestinal microbiota is affected by the aging process with progressive loss of its homeostasis [107]. Senescence leads to a progressive reduction in intestinal microbial biodiversity characterized by a decrease in anti-inflammatory SCFA-producing bacteria and compromised stability of the intestinal microbiota in favor of pathobiont species [108].

Comparing the microbiota of older adults to young adults, it was found that there were lower levels of Firmicutes species (Clostridium cluster and Faecalibacterium prausnitzii) and Actinobacteria (mainly bifidobacteria) and an increase in Proteobacteria populations [109].

Badal et al. [110] highlighted how Akkermansia was the relatively more abundant species in older adults, while Faecalibacterium, Bacteroidaceae, and Lachnospiraceae were relatively reduced.

The gut and brain are closely connected through a bidirectional communication organism known as the brain-gut-microbiota (BGM) axis, through which the brain can influence the composition and function of the gut microbiota [111].

These dysbiotic changes in an aging gut may result in pathogenic alterations of nutrient signaling pathways, stimulation of pro-inflammatory innate immunity, and other pathological conditions [112,113]. Systemic inflammation resulting from intestinal dysbiosis in senescence appears to modulate the neuroinflammatory process, underlying the pathogenesis of cognitive impairment [114]. The increase in the Firmicutes/Bacteroidetes ratio could cause the intestinal accumulation of amyloid precursor proteins from the early stages of Alzheimer’s disease, as well as favor the increase in beta-amyloid at the central level and impairment of spatial learning and memory [115]. It was highlighted how a reduction in anti-inflammatory E. rectale and an increase in pro-inflammatory Escherichia coli/Shigella were present in subjects with cognitive decline and cerebral amyloidosis when compared to healthy control subjects [116].

Some studies investigated the use of probiotics, prebiotics, and synbiotics to improve the biodiversity of the intestinal microbiome while reducing the clinical fragility of older subjects [117,118,119]. The term probiotic refers to all those living microorganisms that confer a benefit to the health of the host when administered in adequate quantities [120]. On the other hand, prebiotics are indigestible food ingredients with a beneficial effect through their selective metabolism in the intestinal tract [121]. The term “symbiotic” is used to describe the synergistic coexistence of both a probiotic and a prebiotic with a final healthy effect superior to that of the probiotic alone [122]. In older adults, symbiotics have been shown to alter the composition of the intestinal microbiota, inducing significant increases in the fecal quantity of bifidobacteria and lactobacilli while helping to protect the IS through the action of the microbiota [123,124].

Refer to Figure 2 for a graphical representation of the impact of aging on vitamin D and microbiome.

## 8. Discussion

Recent literature has shown that the relevant roles of vitamin D and gut microbiota in immune-mediated diseases appear to be strongly interrelated. The role of these factors is also mediated by both the sex and age of the patient.

In the current review, we have seen how vitamin D enhances the activities of the innate IS by controlling the microbiome, especially the intestinal microbiome, and in turn, potentially improves host-microbiota interactions with immune tolerance.

These two factors have different roles depending on age and sex, and both vitamin D and the gut microbiome seem to play a role in the development of immunotolerance, especially at younger ages. This implies that hypovitaminosis D and dysbiosis may be relevant in the genesis of the vast majority of immune-mediated diseases, which develop mainly in younger patients [49,58], while with age they may have a main role as mediators of disease severity, leading the immune system to a more inflammatory phenotype if they are altered [105].

The role of sex in these mechanisms is still unclear, and we believe further research on this subject is needed. However, it remains plausible to think that the sex differences that are witnessed in terms of incidence and clinical course in immune-mediated diseases may be at least partly explained by the differences we have described about vitamin D and the gut microbiome.

Refer to Figure 3 for a graphical representation of the possible pattern of interaction between the different factors considered in this review.

From our narrative review, relevant implications emerge that could have a connotation in the research and the medical approach to these diseases. One aspect that is noteworthy is the possibility of using current knowledge to develop algorithms for risk calculation for each patient. This could also lead to preventive interventions for patients most at risk. For this purpose, the assessment of vitamin D levels, genetic alterations in VDR, and gut microbiome composition could all be relevant aspects of future studies that will need to be weighed against what has emerged and what will emerge concerning the relevance and role of age and sex.

These factors could be important when considering setting specific interventions to improve the overall clinical status of a patient from a prognostic perspective.

For example, these factors could be used in assessing drug resistance in patients and should be interesting to assess precisely (under certain circumstances) whether it is a vitamin D deficiency or dysbiosis as to the reasons why a particular patient is resistant to therapy, as already highlighted in the literature on different diseases [125]. As mentioned, individual factors should be weighed against the age and sex of the patient.

It is clear from current literature that the importance of supplementing patients with appropriate vitamin D dosages, especially when a deficiency is revealed, is important regardless of their sex or age. This is particularly relevant for older adult patients, but it is also true for younger ones [126].

On the other hand, the treatment of dysbiosis remains a topic that needs further study, but hopefully, it will soon become a relevant aspect in research and also in the clinical treatment of patients with immune-mediated diseases.

## 9. Conclusions and Future Perspectives

The importance of these topics and their increased relevance in future clinical applications are testified by the growing literature. This review gives an alternative point of view about immune-mediated diseases, considering the importance of vitamin D and the gut microbiome based on sex differences and the process of aging.

However, there is a demand for more studies related to these matters. More preclinical research is needed for a greater understanding of the molecular mechanisms that hold together vitamin D, gut microbiome, sex, and age to build a translational approach aimed at identifying specific therapeutic interventions.

Also, further clinical research and prospective studies are required to take into account the above-mentioned factors at all stages of the disease, such as incidence, clinical course, and response to medication. In order to define the relevance of interventions in reducing the risk of developing autoimmune or allergic diseases and in improving the clinical course of affected patients, especially those related to the microbiome, more studies are needed.

## 10. Highlights

Vitamin D cooperates with the microbiota in regulating the immune system, improving host-microbiota interactions with immune tolerance as a potential effect;Sex hormones have the ability to influence vitamin D levels and the composition of the intestinal microbiome, promoting the development of gender differences in the presentation of allergic and immune-mediated diseases;The aging process favors the conditions of hypovitaminosis D and dysbiosis, leading the immune system to a more inflammatory phenotype and favoring the development of allergic and immune-mediated diseases.

## Figures and Tables

**Figure 1 biomedicines-12-01023-f001:**
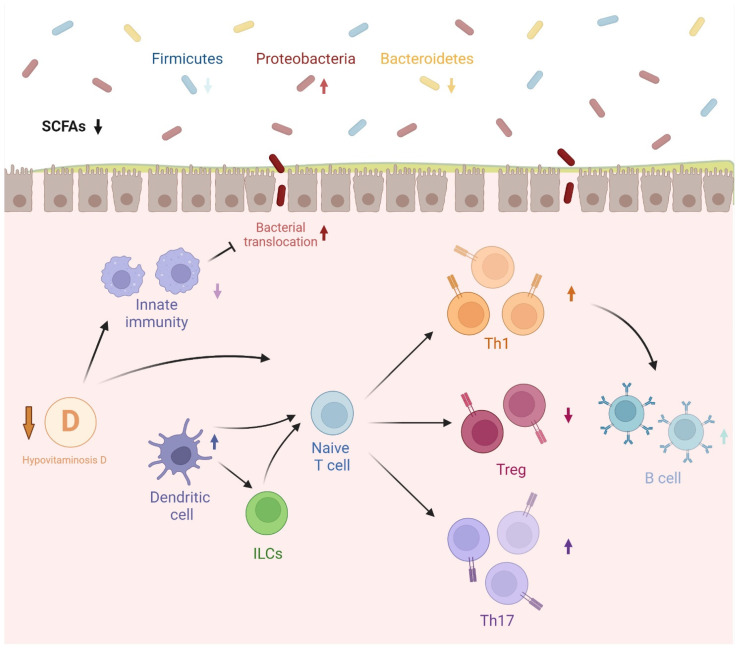
Based on recent studies, we highlight a possible explanatory scheme for the relationship between hypovitaminosis D, dysbiosis, and alterations of the immune system associated with immune-mediated diseases. In particular, a reduction in the level of vitamin D could lead to a reduced immunological tolerance of adaptive immunity and a reduced activity of the innate immune system. The latter aspect may, in turn, lead to less effective control of the microbiome and possible bacterial translocation. Relating instead to the microbiome alone, the most common alterations associated with immune system diseases are described in the image. The arrows stand for the level of activity, or in quantitative terms, relative to the different aspects under consideration. Specifically, if reduced, the arrow is downward, and if increased, it is upward. Abbreviations: ILCs: innate lymphoid cells; Th: T helper; Treg: regulatory T cells; SCFAs: short-chain fatty acids. Created with BioRender.com.

**Figure 2 biomedicines-12-01023-f002:**
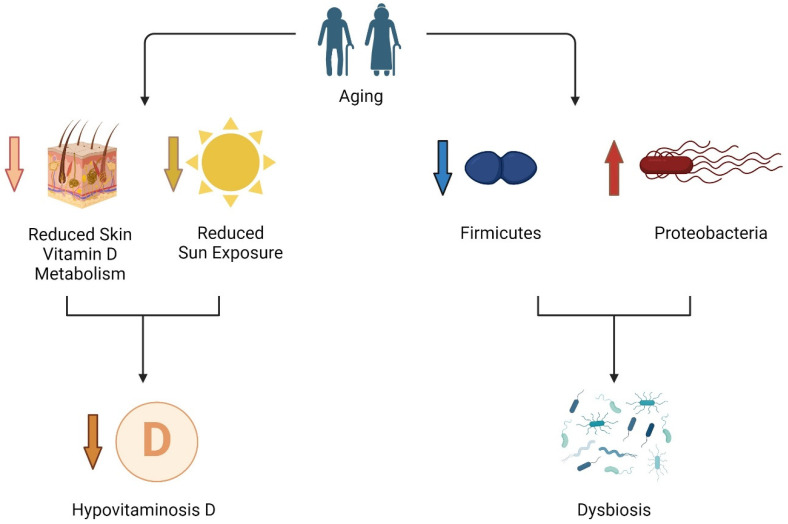
According to current literature, hypovitaminosis D is more frequent in older adults because of the reduction in the capability of our body to synthesize it at skin level, while there is a general reduction in sun exposure, especially in patients in long-term care. At the same time, with aging, there are several changes in the microbiome that could lead to more frequent dysbiosis. Created with BioRender.com.

**Figure 3 biomedicines-12-01023-f003:**
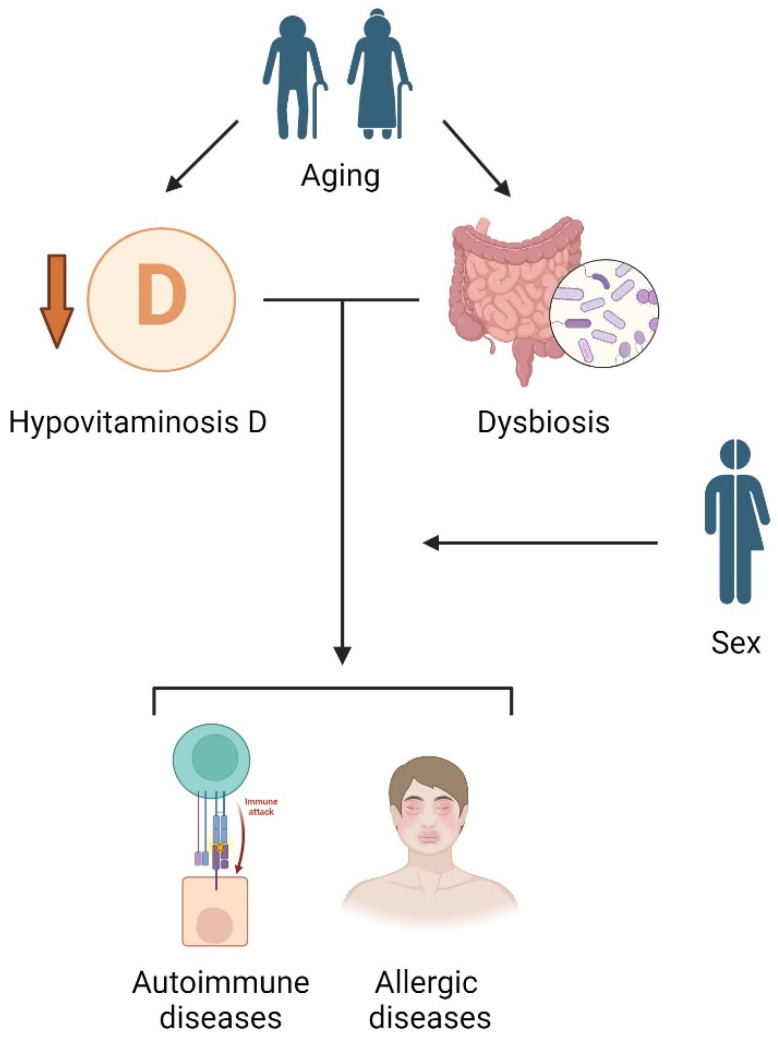
Based on recent literature, we produced a possible explanatory scheme for the relationship between vitamin D, gut microbiome, sex, age, and immune-mediated diseases. The center of the model is characterized by the relevant and partly interdependent role of hypovitaminosis D and dysbiosis in the development of autoimmune or allergic diseases. In this scheme, age is a key aspect because it is a well-known risk factor for developing both vitamin D deficiency and alterations in the gut microbiome, while sex could mediate the effect of the above elements on the incidence and clinical course of immune-mediated diseases, with mechanisms yet to be specified in detail. Created with BioRender.com.

## Data Availability

No new data were created or analyzed in this study. Data sharing is not applicable to this article.

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
