# Peer review of "Gender Differences in the Interplay between Vitamin D and Microbiota in Allergic and Autoimmune Diseases"

_biomedicines, 2024, doi:10.3390/biomedicines12051023_

Round 1

Reviewer 1 Report

Comments and Suggestions for Authors

The topic is of interest and worthy of support. However, there are several deficiencies that have to be corrected. Also the information provided is rather outdated, does not include recent advances in vitamin D (Journal of Investigative Dermatology. 2023, 143(12): 2340-2342) and therefore is not novel and accordingly limited in nature.

When, the authors discuss activation of vitamin D they failed to mention alternative pathways of vitamin D activation by a steroidogenic enzyme CYP11A1 leading to production of several biologically active hydroxyderivatives. This has to be mentioned with appropriate original citations. Note, that CYP11A1 is also expressed in immune cells (Genes Immun 21(3):150-168, 2020. doi: 10.1038/s41435-020-0096-6)

As relates to CYP27B1, for immunoregulation, its experssion in target cells but not kidney is more important.

Inactivation by CYP24A1 is not even mentioned.

As relates to target nuclear receptors the authors list outdated information. Ye, VDR is extremely important but there are important other nuclear receptors for vitamin D3 hydroxyderivatives such as RORalpha and gamma, LXR and Ahr as recently reported. This has to be mentioned with proper referrals to original papers. It must be noted that on some of them 1,25D3 can work as ligand. As related to RORg, it is not a ligand, but alternatively generated 20(OH)D3 and 20,23(OH)2D3 can act as inverse agonists inhibit the downstream IL-17 signaling. Very important information for the topic of this subject. This requires proper corrections, because of the relevance to the subject matter of this review.

Lastly, the readers would appreciate proper mentioning of the diverse role of UVR, of which UVB spectrum generate vitamin D, lumisterol and tachysterol to be activated, among many other factors (Proceedings of the National Academy of Sciences. 2024;121(14):e2308374121. doi: doi:10.1073/pnas.2308374121.)

Author Response

Dear reviewer, many thakns for your suggestions. I have revised the paper accordingly to your indications.

Giuseppe Murdaca

Reviewer 2 Report

Comments and Suggestions for Authors

Dear editor and authors, first of all, I appreciate the possibility of reviewing this work. I think it is a magnificent review, but I notice that there is a lack of a material and methods section that describes the type of bibliographic review carried out, keywords used, databases analyzed, languages, years, etc. Also the type of review, number of articles found, inclusion and exclusion criteria. All these data would give great scientific power to the work and I think they should have done it, but it is not included in the article and I think they should include it.

Author Response

Dear reviewer, many thanks for your suggestions. I have revised the paper accordingly to your indications.

Giuseppe Murdaca

Reviewer 3 Report

Comments and Suggestions for Authors

Major comments:

1. A review article should be easy to read and elegantly combine present knowledge to some new insight. This is not the case with the present manuscript, please substantially rewrite.

2. One figure is by far not enough. Please summarize the information discussed in a number of illustrative figures.

3. The special role of vitamin D is not sufficiently worked out, what are the molecular mechanisms, i.e. which primary vitamin D target genes and regulated pathways are involved. How does this vary between immune cell type and what are the inter individual variations?

Minor comments:

1. Please check for consistent definition and use of abbreviations.

2. The manuscript is not in style of MDPI journals but looks like a transfer from an Elsevier journal.

Comments on the Quality of English Language

Moderate corrections required.

Author Response

(The authors gave the same response as above.)

Reviewer 4 Report

Comments and Suggestions for Authors

The authors have produced a comprehensive review of many published studies relating the variables of vitamin D status, gut microbial composition, sex of participants in these studies and the impact of these factors on immune associated diseases. Mention is made of the decline in vitamin D status with age, but no indication is given to explain this phenomenon. Since in most populations, vitamin D status is determined by the extent of skin exposure to UV radiation from the sun, is the decline in vitamin D status caused by decreased annual exposure to the sun or is it only because of lowering concentrations of 7-dehydrocholesterol in skin cells with age? Some explanation for this observation would be helpful. Is it inevitable that vitamin D status declines with age or is this a consequence of behavioral changes with age?

Comments on the Quality of English Language

Some sentence structure is difficult to follow.

Author Response

(The authors gave the same response as above.)

Round 2

Reviewer 1 Report

Comments and Suggestions for Authors

The authors adequately revised the manuscript

Reviewer 2 Report

Comments and Suggestions for Authors Dear editor, once the authors' response has been analyzed, I believe that the article can be published as a narrative review.

Reviewer 3 Report

Comments and Suggestions for Authors

None

Comments on the Quality of English Language

Moderate corrections needed